# Solar Radiation Modification is projected to increase land carbon storage and to protect the Amazon rainforest

Isobel M Parry<sup>1</sup>, Paul D L Ritchie<sup>1</sup>, Olivier Boucher<sup>2</sup>, Peter M Cox<sup>1,3</sup>, James M Haywood<sup>1,4</sup>, Ulrike Niemeier<sup>5</sup>, Roland Séférian<sup>6</sup>, Simone Tilmes<sup>7</sup>, and Daniele Visioni<sup>8</sup>

**Correspondence:** Isobel M Parry (i.i.parry@exeter.ac.uk)

Abstract. Solar radiation modification (SRM) aims to artificially cool the Earth, counteracting warming from anthropogenic greenhouse gases by increasing the reflection of incoming sunlight. One SRM strategy is stratospheric aerosol injection (SAI), which mimics explosive volcanoes by injecting aerosols into the stratosphere. There are concerns that SAI could suppress vegetation productivity by reducing the amount of sunlight reaching the Earth's surface and by shifting rainfall patterns. Here we examine results from five Earth System Models that use SAI to reduce the global mean temperature from that of a high emissions world (SSP585), to that of a more moderate global warming scenario (SSP245). Compared to SSP245, the SAI simulations project higher global net primary productivity (NPP) values (+15.6%) and higher land carbon storage (+5.9%), primarily because of increased CO<sub>2</sub> fertilisation. The effects of SAI are especially clear in Amazonia where land carbon storage increases under G6suplhur compared to both SSP245 (+8.6%) and SSP585 (+10.8%), even though the latter scenario has the same atmospheric CO<sub>2</sub> scenario as G6sulfur. Our results therefore suggest that SAI could provide some protection against the risk of climate change induced carbon losses from the Amazon rainforest.

## 1 Introduction

The 2015 Paris Climate Agreement set out to hold global warming to well below 2°C, and to pursue efforts to limit global warming to 1.5°C above pre-industrial levels (Royal Society (Great Britain), 2009). However, global anthropogenic CO<sub>2</sub> emissions - the primary cause of global warming - have continued to rise, reaching new record highs in 2023 and 2024. The global temperature for 2023 was between 1.34 and 1.54°C warmer than preindustrial (Witze, 2024), while 2024 had a global warming of 1.55 °C (+/- 0.13 °C)(World Meteorological Organization, 2025). The Paris targets apply to decade-long warming, but even then, it is estimated that a little more than a decade more of current global CO<sub>2</sub> emissions would take decadal mean warm-

<sup>&</sup>lt;sup>1</sup>Department of Mathematics and Statistics, Faculty of Environment, Science and Economy, University of Exeter, Exeter, EX4 40E, UK

<sup>&</sup>lt;sup>2</sup>Institut Pierre-Simon Laplace, Sorbonne Université / CNRS, Paris, France

<sup>&</sup>lt;sup>3</sup>Global Systems Institute, University of Exeter, Exeter, UK

<sup>&</sup>lt;sup>4</sup>Met Office Hadley Centre, Exeter, UK

<sup>&</sup>lt;sup>5</sup>Max Planck Institute for Meteorology, Hamburg, Germany

<sup>&</sup>lt;sup>6</sup>CNRM, Université de Toulouse, Météo-France, CNRS, Toulouse, France

<sup>&</sup>lt;sup>7</sup>Atmospheric Chemistry, Observations and Modeling Laboratory, National Center for Atmospheric Research, Boulder, CO, USA

<sup>&</sup>lt;sup>8</sup>Sibley School for Mechanical and Aerospace Engineering, Cornell University, Ithaca, NY, USA

50

ing over 1.5°C (Cox et al., 2024; Betts et al., 2023). Avoiding 1.5°C of global warming therefore looks increasingly unlikely through conventional mitigation, even if carbon dioxide removal measures are also employed at scale.

Further global warming also increases the risks associated with potential climate tipping points, such as ice-sheet melt, collapse of the ocean's thermohaline circulation, and Amazon Forest dieback (McKay et al., 2022; Parry et al., 2022). Climate tipping points are described as conditions beyond which the changes in a part of the climate system become self-perpetuating, often with abrupt and significant impact (McKay et al., 2022). In many cases, there may be early warning signals for these tipping points, and for slower tipping elements such as ice sheet melt, there may be sufficient time to avoid tipping through the management of greenhouse gases alone (Ritchie et al., 2021). However, for faster tipping elements such as forest dieback, a faster acting 'emergency' measure would be required to avoid tipping.

The failure of humanity to slow global warming, and reduce the associated tipping point risks, has led to a surge in research around Solar Radiation Modification (SRM). SRM aims to artificially cool the planet by increasing the reflection of incoming sunlight back into space through various means (Irvine et al., 2016). Stratospheric aerosol injection (SAI) is the most prominent SRM approach. This aims to mimic the cooling effect of explosive volcanoes by injecting reflective aerosols, or their gaseous precursors, into the stratosphere (Crutzen, 2006; Robock et al., 2009). SAI has the potential to rapidly cool the planet, but there are numerous concerns about this approach associated with governance, equity, moral hazard, and unintended consequences (Robock, 2020; Abiodun et al., 2021; Pope et al., 2012; Trisos et al., 2018; Barrett, 2008; Haywood et al., 2022a). SRM is also a topic that elicits very strong responses, which has tended to marginalise SRM research, at a time when it needs to be discussed more widely and openly (Wieners et al., 2023).

Here we address questions about the potential impact of SAI on global vegetation productivity, land carbon storage (the storage of carbon in terrestrial ecosystems, in vegetation, soil and litter) and the risk of climate-driven Amazon Forest carbon loss (Cox et al., 2004; Trisos et al., 2018; Barrett, 2008). Both land carbon storage and net primary productivity (NPP - the rate at which carbon is gained via photosynthesis minus respiratory losses) are often used as proxies for evaluating vegetation health. NPP reflects regional carbon sequestration via photosynthesis, and land carbon storage is directly impacted by carbon stored within vegetation. We use results from recent Geoengineering Model Intercomparison Project (GeoMIP) projections (Kravitz et al., 2015; Visioni et al., 2021) carried out with five of the latest Coupled Model Intercomparison Project Phase 6 (CMIP6) (Eyring et al., 2016) generation of Earth Systems Models (ESMs). Instances of Amazon forest dieback in CMIP6 models are typically associated with declining net primary productivity (NPP) (Cox et al., 2004), we therefore look specifically at how SAI may affect the change in NPP in Amazonia. These ESMs model the responses of global vegetation and land carbon storage to changes in climate and atmospheric CO<sub>2</sub>. All of these models include some parametrisation of nutrient constraints on CO<sub>2</sub> fertilisation of photosynthesis (Norby, 2011; Terrer et al., 2018), and therefore on the resulting increases in NPP and land carbon storage. Three have an explicit nitrogen cycle, while the other two represent implicit nutrient limitations (Arora et al., 2020).

We examine the projected climate under SAI (G6sulfur), which aims to reduce the forcing from the high emissions scenario (SSP585) to the medium forcing scenario (SSP245) using equatorial SO<sub>2</sub> injections into the stratosphere. We also compare to a medium emissions scenario (SSP245) which reaches a similar global warming to G6sulfur, but instead through reduced

60

greenhouse gas concentrations rather than SAI. Therefore, the comparison between G6sulfur and SSP245 allows us to compare the impacts of SAI to the impacts of a similar level of global warming achieved through conventional mitigation. Since G6sulfur applies the higher CO<sub>2</sub> scenario of SSP585, G6Sulpur will benefit from higher CO<sub>2</sub> fertilization. By contrast, the comparison between G6sulfur and SSP585 isolates the climatic impacts of SAI. To represent the uncertainties in the model projections we present our results as a likely range (plus and minus one standard deviation) around an ensemble mean as well as the individual ESM projections.

Previous single-model studies have explored changes in NPP under SRM scenarios (Xia et al., 2015; Dagon and Schrag, 2019; Yang et al., 2020; Lauvset et al., 2017; Tjiputra et al., 2016; Zhao and Cao, 2022; Jones et al., 2011; Duan et al., 2020). Multi-model simulations include Kravitz et al. (2013, 2021); Glienke et al. (2015); Plazzotta et al. (2019), however, these studies assumed that the impact of SAI is equivalent to turning down the solar constant (GeoMIP G1); they did not explicitly represent the transport, and chemical evolution of sulphate aerosols. A multi-model study of the impacts of explicit SAI injection was carried out using the previous generation of GeoMIP G4 simulations (Plazzotta et al., 2018) but did not describe the simulated responses of vegetation. Here, we use the GeoMIP G6 simulations carried out with the more recent CMIP6 ESMs. We focus specifically on the potential impacts of SRM on the Amazon rainforest, which is one of the most diverse ecosystems on Earth and may be vulnerable to both CO<sub>2</sub>-induced climate change (Cox et al., 2004) and shifting rainfall patterns due to changing aerosol loadings (Jones et al., 2013; Cox et al., 2008). Key questions that we address include: (i) how will the reduced sunlight at the surface affect vegetation productivity; and (ii) how will any residual regional climate change impact vegetation?

### 2 Methods

## 2.1 CMIP6 models, experiment runs, and data used

Our study focuses on results from Earth System Model simulations from Phase 6 of the Coupled Model Intercomparison Project Phase 6 (CMIP6)(Eyring et al., 2016) for three scenarios, namely SSP245, SSP585 and G6sulfur (Kravitz et al., 2015; O'Neill et al., 2016). Broadly, SSP585 is a fossil fuel intensive high emissions scenario, while SSP245 is a medium radiative forcing scenario with medium challenges to conventional mitigation, which assumes that historical trends continue without substantial deviations (O'Neill et al., 2016). Both scenarios run from 2015 to 2100. G6sulfur is part of the Geoengineering Model Intercomparison Project (GeoMIP). G6sulfur sets out to reduce the forcing from the high forcing scenario (SSP585) to the medium forcing scenario (SSP245) using stratospheric sulphate aerosols injected in a line from 10°S to 10°N along a single longitude band (0°), starting in 2020 and continuing to 2100. The amount of sulphate injection required to reduce radiative forcing to that of the medium forcing scenario SSP245 varies between models and the necessary amount is calculated on an individual model basis (Kravitz et al., 2015). Table 2 highlights differences between how individual models simulate G6sulfur. All three scenarios were concatenated onto historical runs (1850-2014) for each model. G6sulfur simulations begin in 2020, and so for the missing five years from 2015-2019 SSP585 data is used to bridge the gap.

**Table 1.** Details of G6sulfur experiment setup within the models analysed in this work. \* Injected at the Equator at 25 km in deviation from the protocol described by Kravitz et al. (2015) (Kravitz et al., 2015; Visioni et al., 2021).

| Model                                  | G6sulfur simulation length | Stratospheric<br>aerosols in<br>G6sulfur                       | Adjustment interval (Tilmes et al., 2022)                                   | Interactive strato-<br>spheric ozone (Vi-<br>sioni et al., 2021) |
|----------------------------------------|----------------------------|----------------------------------------------------------------|-----------------------------------------------------------------------------|------------------------------------------------------------------|
| CESM2-WACCM (Danabasoglu et al., 2020) | 2020–2100                  | From SO <sub>2</sub> injection*                                | Yearly via feed-<br>back control<br>algorithm (Mac-<br>Martin et al., 2017) | Yes                                                              |
| CNRM-ESM1-2 (Séférian et al., 2019)    | 2015–2100                  | Aerosol optical depth scaled from (Tilmes et al., 2015)        | Prescribed in (Tilmes et al., 2015)                                         | Yes                                                              |
| IPSL-CM6A-LR (Boucher et al., 2020)    | 2020–2100                  | From SO <sub>2</sub> injection                                 | Decadal stepping of injections                                              | No                                                               |
| MPI-ESM1-2-LR (Mauritsen et al., 2019) | 2015–2100                  | Aerosol optical depth scaled from (Niemeier and Schmidt, 2017) | Prescribed in (Niemeier and Schmidt, 2017)                                  | No                                                               |
| UKESM1-0-LL (Sellar et al., 2019)      | 2020–2100                  | From SO <sub>2</sub> injection                                 | Decadal stepping of injections                                              | Yes                                                              |

For this study, single ensemble members from five climate models (provided in Table 2) that performed the G6sulfur CMIP6 simulation are used (Meehl et al., 2014). We examine the model output diagnostics for terrestrial carbon storage (cLand) and net primary productivity (NPP) to assess the impact of the SAI (G6sulfur) GeoMIP experiment on the terrestrial carbon cycle. Four of the models used in this study show good agreement with observations for land ecosystems and the carbon cycle according to the IPCC AR6 report (the fifth, CNRM-ESM1-2, was not assessed in this chapter of the AR6 (Canadell et al., 2023)). We also analyse surface temperature and precipitation differences as these are key for explaining the differences across the scenarios. All data from the CMIP6 experiments were first linearly interpolated onto a world grid with 1° x 1° resolution. Analysis of the Amazon used a region defined by the latitudes 20°S to 14°N and longitudes 83°W to 34°W.

# 2.2 Validation of models against volcanic eruptions

Although it is impossible to evaluate SRM runs against observations, we can use volcanic eruptions as a proxy. We see consistent spatial patterns between models in response to the volcanic eruptions of Pinatubo, El Chichon, Gunung Agung, and Krakatoa (Figure 1), which are comparable to those observed in atmospheric inversion models (Piao et al., 2020). Of all the

**Figure 1.** (a-e) maps showing the anomaly in the land carbon flux (gC m<sup>-2</sup> yr<sup>-1</sup>) due to volcanoes for five models. These display average anomaly of the land carbon flux for the average of the first two years following the eruption of 1991 Pinatubo, 1982 El Chichon, 1963 Gunung Agung and 1883 Krakatoa eruptions, compared to the 5 years preceding these eruptions. (**f**) map showing the ensemble mean of the five models.

**Table 2.** The 5 CMIP6 Earth System models included in this study and relevant features of their land carbon cycle components (Arora et al., 2020). \*No nitrogen cycle is included, but there is a parametrisation to limit CO<sub>2</sub> fertilisation at high concentrations of CO<sub>2</sub> (Boucher et al., 2020).

| Model                                  | Institution | <b>Land Surface Model</b> | Nitrogen cycle                         | Fire |
|----------------------------------------|-------------|---------------------------|----------------------------------------|------|
| CESM2-WACCM (Danabasoglu et al., 2020) | NCAR        | CLM5                      | Yes                                    | Yes  |
| CNRM-ESM1-2 (Séférian et al., 2019)    | CNRM        | ISBA-CTRIP                | No (implicit, derived from (Yin, 2002) | Yes  |
| IPSL-CM6A-LR (Boucher et al., 2020)    | IPSL        | ORCHIDEE branch 2.0       | No*                                    | No   |
| MPI-ESM1-2-LR (Mauritsen et al., 2019) | MPI         | JSBACH 3.2                | Yes                                    | Yes  |
| UKESM1-0-LL(Sellar et al., 2019)       | UK          | JULES-ES-1.0              | Yes                                    | No   |

Figure 2. Timeseries showing the evolution of the cumulative land carbon uptake anomaly (relative to the average of the five years before the eruption) in the five years before and after a large volcanic eruption. Four volcanic eruptions are considered here: Pinatubo (1991), El Chichon (1982), Gunung Agung (1963), and Krakatoa (1883). The bold black line represents the ensemble mean for all the models and eruptions analysed here while the bold grey line represents the ensemble mean of the global carbon budget data for the period around the four eruptions.

models, MPI-ESM1-2-LR experiences the greatest area of notable decreases in the land carbon sink, though similar decreases are also observable in CNRM-ESM2-1, IPSL-CM6A-LR and UKESM1-0-LL. Meanwhile, UKESM1-0-LL appears to have the largest areas of notable increase in the land carbon sink due to volcanic eruptions, with these increases especially prominent in southern regions of the African continent. Observed responses to individual volcanic eruptions differ due to changes in the ENSO phase (Jones et al., 2001), which are often different to the ENSO phase being experienced in the models for the same year. Despite this, we observe an increase in land carbon uptake after an eruption in both models and the global carbon budget

(GCB) data, though the models appear to underestimate the increase in the land carbon sink compared with GCB data from 2 years post-eruption onwards (Figure 2). On this basis, we conclude that the ESMs have a qualitatively realistic response to stratospheric aerosol injection by volcanoes (with the global land carbon sink tending to increase after explosive volcanic eruptions), but that the magnitude of this response appears to be slightly underestimated compared to the GCB data. This may be because this generation of ESMs does not routinely model the enhancement of photosynthesis by diffuse radiation (Mercado et al., 2009). As a result, the results presented in this study are likely to be conservative with respect to the positive impacts of SAI on NPP and land carbon storage.

## 3 Results and Discussion

Figure 3 shows the changes in global mean (relative to the pre-industrial reference period, 1850-1899) for four variables (surface temperature, precipitation, land net primary productivity, and land carbon storage). On each panel there are lines representing the two conventional climate change scenarios (SSP245 and SSP585) and the SRM scenario (G6sulfur). All scenarios exhibit rising temperatures compared with pre-industrial temperatures, though as expected, the SSP585 scenario shows the greatest increase reaching an ensemble-mean global warming of approximately 6°C by 2100. By design, the G6sulfur profile tracks that of SSP245 reaching about 3.5°C of global warming by the end of the century in the ensemble mean. Increases relative to the pre-industrial period are observed in precipitation, NPP and land carbon storage with scenarios starting to noticeably diverge after 2025. Precipitation increase in G6sulfur plateaus around 2050, then decreases to only about 1.5% greater than pre-industrial levels. Increases in NPP and land carbon storage are observed despite an expected increase in respiration resulting from increased temperatures (Ballantyne et al., 2017; Bond-Lamberty and Thomson, 2010). The larger increase in land carbon storage in G6sulfur results from both the positive effects of additional CO<sub>2</sub> (like SSP585) but without the extra soil and plant respiration that arises from the additional warming of SSP585 relative to SSP245. Respiration increases under warming due to the temperature dependence of the reaction rates of enzymes involved in respiratory pathways (Heskel et al., 2016; Chen et al., 2024). Lower temperatures therefore reduce respiration rates and increase NPP, which results in greater increases in land carbon storage.

The range in land carbon storage results from differences in land carbon process representation across the models. Models with an explicit nitrogen cycle tend to have lower land carbon uptake than models that do not (Arora et al., 2020), as nitrogen limitations typically suppress CO<sub>2</sub> fertilisation (Terrer et al., 2018). However, the extent of nutrient limitation remains one of the greatest sources of uncertainty regarding the magnitude of CO<sub>2</sub> fertilisation (Terrer et al., 2018). CNRM-ESM2-1, which lacks an explicit representation of the nitrogen cycle (see Table 2), shows notably greater uptake than the other models (Supplementary Figure S1). This model also has a high residence timescale for land carbon, which contributes to its elevated land carbon uptake (Davies-Barnard et al., 2020; Arora et al., 2020).

Figure 4 depicts differences between the SSP585 and G6sulfur scenarios by the end of the 21st century, where the hatching in these plots removes changes in land carbon storage that are not due to SAI from our analysis. See Supplementary Figure S2 for prescribed forest fraction changes between SSP585 and SSP245. In Figure 4a there is an average global temperature

**Figure 3.** Timeseries showing the evolution of the decadal means of **(a)** surface temperature, **(b)** precipitation, **(c)** net primary productivity, and **(d)** land carbon storage anomalies relative to the pre-industrial period (1850-1900) with time from 1900 to 2100. Solid curves represent the ensemble mean of the five CMIP6 models (see Table 2), while the banding represents one standard deviation from this mean.

decrease of approximately 2.2°C in G6sulfur relative to SSP585, although this is not homogenous across the globe. CESM2-WACCM, IPSL-CM6A-LR and UKESM1-0-LL all experience similar levels of global cooling, between 2.5°C and 2.56°C (Figure 5). CNRM-ESM1-2 meanwhile experiences less cooling, with a global average of 1.86°C. Nevertheless, the level of cooling in the northern polar regions is similar to the cooling in CESM2-WACCM, IPSL-CM6A-LR and UKESM1-0-LL, because regions with less cooling, or even warming, are concentrated outside of the northern poles. MPI-ESM1-2-LR shows notably less cooling than even CNRM-ESM2-1 with global mean temperatures only 1.38°C cooler, while the northern polar regions in MPI-ESM1-LR show little cooling, and warming of up to 2°C in some regions. In general, the northern hemisphere cools considerably more than the southern, with polar regions in particular experiencing a cooling of 5°C or more, while much of the ocean in the southern hemisphere only experiences decreases between 1 and 2°C, a trend observed in most models (compare Figures 4 and 5).

SRM affects precipitation patterns compared to SSP585 (Figure 4b) resulting in an average global decrease of about 6.4%. The most pronounced changes occur around the equator, due to the response of the Intertropical Convergence Zone (ITCZ) to uneven cooling between the northern and southern hemispheres (Frierson and Hwang, 2012; Donohoe et al., 2013). These shifts in the ITCZ are more obvious when observing precipitation changes in individual models (Figure 6) as the location of the ITCZ differs between models. UKESM1-0-LL experiences the greatest drying of the models analysed in this study, on average drying by 0.26 mm day<sup>-1</sup> globally. Meanwhile, MPI-ESM1-2-LR shows the smallest average global decrease in precipitation

**Figure 4.** Maps showing the difference between the 2090-2100 means of G6sulfur and SSP585 for (a) surface temperature (°C), (b) precipitation (mm day<sup>-1</sup>), (c) net primary productivity (kgC m<sup>-2</sup> yr<sup>-1</sup>), and (d) land carbon storage (kgC m<sup>-2</sup>). The numbers shown in the titles give the global average (a, b) or global land average for each map (c, d). Stippling indicates regions where the standard deviation across the models is more than the ensemble mean change, i.e. the coefficient of variation is more than 1. Hatching indicates where the forest fraction as prescribed in the SSP245 and SSP585 land-use scenarios, differs by more than 0.1 (on which the scenario for G6sulfur is based) land-use scenarios, differs by greater than 0.1. In these regions, differences in regional climate, NPP and land carbon may arise due to the different land-use scenario, rather than the impacts of SRM. The hatching is used to indicated areas which we therefore remove from our comparative analyses.

of 0.157 mm day<sup>-1</sup>. While this model still shows large decreases in precipitation around the equator, the mid to high latitudes show a notably smaller decrease in precipitation than the other models analysed here. All models show large decreases in precipitation over Indonesia of more than 1 mm day<sup>-1</sup>, while all models except CNRM-ESM2-1 show notable decreases in precipitation over central areas of the African continent (Figure 6). Going back to the ensemble mean (Figure 4b), the effect of SAI is especially strong over the equatorial oceans, with smaller decreases spread across the mid to high-latitudes. Conversely, a statistically significant increase is observed over eastern Australia, where vegetation is largely water-limited (Chen et al., 2022). This region also shows a notable rise in NPP and land carbon storage (i.e. a greening under the G6sulfur scenario, Figure 4c, d).

Relative to SSP585, global mean NPP increases in three models in G6sulfur (Figure 7). There is a global mean decrease of NPP in IPSL-CM6A-LR and MPI-ESM1-2-LR, associated with reduced precipitation, which can reduce NPP in water limited

**Figure 5.** Maps showing the difference between the 2090-2100 means of G6sulfur and SSP585 for surface temperature for individual models. The numbers shown in the titles give the global average. Hatching indicates where differences in the forest fraction as prescribed by the land use scenarios between SSP245 and SSP585 is greater than 0.1 (see Figure S2).

**Figure 6.** Maps showing the difference between the 2090-2100 means of G6sulfur and SSP585 for precipitation (mm day<sup>-1</sup>) for individual models. The numbers shown in the titles give the global average. Hatching indicates where differences in the forest fraction as prescribed by the land use scenarios between SSP245 and SSP585 is greater than 0.1 (see Figure S2).

**Figure 7.** Maps showing the difference between the 2090-2100 means of G6sulfur and SSP585 for net primary productivity (kgC m<sup>-2</sup> yr<sup>-1</sup>) for individual models. The numbers shown in the titles give the global land average. Hatching indicates where differences in the forest fraction as prescribed by the land use scenarios between SSP245 and SSP585 (on which the scenario for G6sulfur is based) is greater than 0.1 (see Figure S3). In these regions, differences in regional climate, NPP and land carbon may arise due to the different land-use scenario, rather than the impacts of SRM. The hatching is used to indicated areas which we therefore remove from our comparative analyses.

180

regions, and insufficient cooling, which can lead to plant temperatures going beyond the optimum for photosynthesis (see Supplementary Table S1). We also observe decreases in NPP (up to 0.2 kgCm<sup>-2</sup>yr<sup>-1</sup>) over large areas of the northern high latitudes in CESM2-WACCM where the model has the greatest cooling, though this is not visible in the global average due to counteracting increases in the southern hemisphere.

Global land carbon storage increases in all models except IPSL-CM6A-LR (Figure 8), which shows some areas of decrease in northern high latitudes and eastern Africa with little offset from land carbon storage increases in other regions. The only other model with a notable decrease in land carbon storage is MPI-ESM1-2-LR which has a region of decrease of more than 5 kgC m<sup>-2</sup> in central regions of Africa. UKESM1-0-LL shows the greatest increase in land carbon storage of the models analysed, with a global average increase of 0.658 kgC m<sup>-2</sup>. These increases are concentrated around Amazonia, central Africa and south-east Asia, with smaller increases occurring around the mid to high latitudes.

Tropical forests are sensitive to climate changes. Of particular concern is drying in Amazonia (Parry et al., 2022). Increases in temperature above optimum levels also result in reduced photosynthesis, and therefore reduced NPP, in most current models (Doughty and Goulden, 2009). There is observational evidence of partial acclimation of photosynthesis to warming, although this is not yet fully represented in Earth System Models (Kumarathunge et al., 2019). By contrast, reducing temperatures via SAI bring tropical forests closer to their optimum photosynthesis temperatures while also reducing respiration rates, thereby increasing both NPP and land carbon storage (Ballantyne et al., 2017; Bond-Lamberty and Thomson, 2010).

The Amazon rainforest is projected to benefit from G6sulfur compared to SSP585, with an ensemble-mean increase in NPP of about 10.8%. This corresponds to a positive impact on land carbon storage in the Amazon (+7.1%) with increases observed throughout Amazonia (Figure 9), despite localised areas of reduced rainfall in the east and west. However, the area of greatest decrease in the ensemble mean precipitation occurs on the north-western coast of South America (a decrease of more than 1mm day<sup>-1</sup> and corresponds to a decrease in NPP in the model ensemble mean between 0.1 and 0.2 kgC m<sup>-2</sup> yr<sup>-1</sup> (Figure 10 and Figure 6). This is in contrast to the decreases seen in NPP and vegetation carbon in idealised scenarios (Xia et al., 2015) and in a single climate model (Yang et al., 2020; Tjiputra et al., 2016). Four of the five ESMs project increases in land carbon storage across Amazonia due to SRM. The fifth model (MPI-ESM1-2-LR) also projects major increases in land carbon storage in the north and east of Amazonia, but this is partially counteracted by a band of reduced vegetation carbon in the southwest (Figure 9d). Overall though, all models project increases in Amazonian land carbon storage (ranging from 6 to 41 PgC) relative to SSP585 (Supplementary Table S1). As the SSP585 and G6sulfur scenarios have the same prescribed evolution of atmospheric CO<sub>2</sub>, these increases are solely due to the impact of SAI on the Amazonian climate.

These increases in NPP and land carbon storage suggest that SRM would offer protection against the risk of climate driven carbon loss in the Amazon forest (Flores et al., 2024). Other tropical forest regions, where precipitation declines are larger under SRM, are not afforded the same degree of protection via increases in NPP. We have previously seen evidence of localised Amazon forest dieback, in multiple models, in idealised 1% CO<sub>2</sub> per year runs (Parry et al., 2022), although only one of the models studied here (IPSL-CM6A-LR) exhibits dieback within the 21<sup>st</sup> century under SSP585. As forest dieback arises when the rate of forest carbon loss exceeds the rate of carbon gain, changes in simulated NPP imply changes in the risk of forest dieback. However, while the results presented here indicate that SRM would protect against carbon losses in the Amazon forest,

**Figure 8.** Maps showing the difference between the 2090-2100 means of G6sulfur and SSP585 for land carbon (kgC m-2) for individual models. The numbers shown in the titles give the global land average. Hatching indicates where differences in the forest fraction as prescribed by the land use scenarios between SSP245 and SSP585 is greater than 0.1 (see Figure S2).

**Figure 9.** Maps for individual models (**a-e**) and the model ensemble mean (**f**) showing the difference between the 2090-2100 means of G6sulfur and SSP585 for land carbon storage (kgC m<sup>-2</sup>) in Amazonia. The numbers shown in the titles give the regional land average for each map. Stippling in (**f**) indicates regions where the standard deviation across the models is more than the ensemble mean change, i.e. the coefficient of variation is more than 1. Hatching indicates where the forest fraction as prescribed in the SSP245 and SSP585 land-use scenarios, differs by more than 0.1(on which the scenario for G6sulfur is based) land-use scenarios, differs by greater than 0.1. In these regions, differences in regional climate, NPP and land carbon may arise due to the different land-use scenario, rather than the impacts of SRM. The hatching is used to indicated areas which we therefore remove from our comparative analyses.

in the future it would be beneficial to run SRM experiments over longer periods that yield clearer examples of forest dieback in the absence of SRM. Though the Amazon experiences relatively small decreases in precipitation in most models, other tropical forest regions experience larger decreases in precipitation (Figure 6). As a result, the increases in NPP and land carbon storage in these areas is less pronounced than is modelled for Amazonia (Figure 7 and Figure 8).

The scenarios SSP245 and G6sulfur have similar global mean temperatures, but there are higher atmospheric CO<sub>2</sub> concentrations under G6sulfur. This allows us to compare the relative impact of a given global warming achieved through conventional mitigation (SSP245) with that achieved through SAI (G6sulfur). Figure 11 shows equivalent maps to Figure 4, but now for G6sulfur relative to SSP245. By design, the projected global warming is very similar. However, there is a residual warming of the polar regions in G6sulfur(Visioni et al., 2021). This is most pronounced in MPI-ESM1-2-LR, which experienced increases of more than 4°C in some northern polar regions, and is prominent in IPSL-CM6A-LR and UKESM1-0-LL (Figure 12). The simple strategy of injection at the Equator has been modified in other studies to reduce these residual impacts (Kravitz et al.,

**Figure 10.** Maps showing the difference between the 2090-2100 means of G6sulfur and SSP585 in the Amazon for (a) surface temperature (°C), (b) precipitation (mm day<sup>-1</sup>), (c) net primary productivity (kgC m<sup>-2</sup> yr<sup>-1</sup>, and (d) land carbon (kgC m<sup>-2</sup>). The numbers shown in the titles give the average of the displayed area (a, b) or land average of the displayed area for each map (c, d). Stippling indicates regions where the standard deviation across the models is more than the ensemble mean change while hatching indicates where differences in the forest fraction as prescribed by the land use scenarios between SSP245 and SSP585 is greater than 0.1 (see Figure S2).

2017). IPSL-CM6A-LR produces the closest average global temperature in G6sulfur compared to SSP245 with a difference of only 0.016°C. As this value is a global mean there are of course regional variations in the level of cooling achieved, with residual warming still being present in northern high latitudes in this model. The greatest difference between SSP245 and G6sulfur global average temperatures is observed in CNRM-ESM2-1 in which G6sulfur is 0.275°C warmer than SSP245 by the end of the century, with lots of this warming concentrated over land masses. In particular, Amazonia and northern regions of Russia experience the largest area of warming between 2°C and 3°C compared to SSP245.

Global mean precipitation decreases slightly on average across the models (-3.7%) (Figure 11b), mainly due to a band of decreased precipitation around the Equator, which is associated with model specific shifts in the ITCZ (Figure 13). Such regional shifts in rainfall can have major regional impacts, an important consideration for SRM (Zhao and Cao, 2022). A statistically significant (clear of stippling) increase in precipitation is also observed in eastern Australia, corresponding to increases in NPP and land carbon storage (Figure 11c, d). Figure 13 shows that IPSL-CM6A-LR and UKESM1-0-LL have

**Figure 11.** Maps showing the difference between the 2090-2100 means of G6sulfur and SSP245 for (**a**) surface temperature (°C), (**b**) precipitation (mm day<sup>-1</sup>), (**c**) net primary productivity (kgC m<sup>-2</sup> yr<sup>-1</sup>), and (**d**) land carbon storage (kgC m<sup>-2</sup>). The numbers shown in the titles give the global average (**a**, **b**) or global land average for each map (**c**, **d**). Stippling indicates regions where the standard deviation across the models is more than the ensemble mean change. Hatching indicates where the forest fraction as prescribed in the SSP245 and SSP585 land-use scenarios, differs by more than 0.1. (on which the scenario for G6sulfur is based) land-use scenarios, differs by greater than 0.1. In these regions, differences in regional climate, NPP and land carbon may arise due to the different land-use scenario, rather than the impacts of SRM. The hatching is used to indicated areas which we therefore remove from our comparative analyses.

the greatest average decrease in precipitation globally in G6sulfur compared to SSP245. While UKESM1-0-LL appears to have more areas where precipitation decreases more than 1mm day<sup>-1</sup>, it also has more areas in which precipitation increases notably, thereby offsetting the areas of decrease when the average is taken. Similarly, MPI-ESM1-2-LR has large areas that experience a decrease in precipitation of more than 1mm day<sup>-1</sup>, but this is not necessarily reflected in the global average due to a substantial number of areas in which precipitation increases more than 0.6mm day<sup>-1</sup>. In most models, large decreases in precipitation are concentrated in a band around the equator, with larger increases in precipitation similarly found in this region. Smaller changes in precipitation tend to occur in the mid-latitudes of the models.

Figure 11c depicts an increase in global mean NPP of about 15.6% in G6sulfur compared to SSP245. Generally, areas with increased NPP in the ensemble mean coincide with agreement between models on the projected sign for this change. Much of the land in the southern hemisphere, southern Asia, and central America shows NPP increases in G6sulfur relative to SSP245. The larger NPP increases in G6sulfur, which are primarily due to additional CO<sub>2</sub> fertilisation, drive a larger land carbon

**Figure 12.** Maps showing the difference between the 2090-2100 means of G6sulfur and SSP245 for surface temperature for individual models. The numbers shown in the titles give the global average. Hatching indicates where differences in the forest fraction as prescribed by the land use scenarios between SSP245 and SSP585 is greater than 0.1 (see Figure S2).

**Figure 13.** Maps showing the difference between the 2090-2100 means of G6sulfur and SSP245 for precipitation (mm day<sup>-1</sup>) for individual models. The numbers shown in the titles give the global average. Hatching indicates where differences in the forest fraction as prescribed by the land use scenarios between SSP245 and SSP585 is greater than 0.1 (see Figure S2).

storage increase compared to SSP245 (+5.9% as opposed to +2.8%). The largest increases in land carbon storage are observed in Indonesia, a difference that is not observable in the comparison between G6sulfur and SSP585. All models show global mean increases in both NPP and land carbon storage in G6sulfur compared to SSP245 (Figure 14 and Figure 15). CESM2-WACCM shows some decreases in NPP and land carbon storage in areas of eastern Europe, while IPSL-CM6A-LR has decreases of a similar scale in NPP and land carbon storage in northern Amazonia, Indonesia and central areas of the African continent. The increase in land carbon storage observed in IPSL-CM6A-LR, while the smallest of all the models, contrasts with the decrease in land carbon storage observed in the same model for the comparison with SSP585. All other models display the same global upward trend in land carbon storage across both the SSP585 and SSP245 comparison. UKESM1-0-LL has the greatest average increase in NPP of the models analysed (0.0748 kgC m<sup>-2</sup> yr<sup>-1</sup>) compared to SSP245. Notable increases in NPP are largely focused around the equator and at lower latitudes, something reflected in the land carbon storage increases in UKESM1-0-LL. Conversely, the model which experiences the greatest increase in land carbon storage is CNRM-ESM2-1, which shows notable increases in land carbon storage across the globe, with the largest increases in the northern mid to high latitudes (Figure 14 and Figure 15).

Once again, the Amazon rainforest demonstrates enhanced carbon storage (+8.6%) and NPP (+13.8%) under G6sulfur compared to SSP245 (Figure 16), primarily due to the CO<sub>2</sub> fertilisation effect, where an increase in photosynthesis results in an increase in NPP and biomass (Cox et al., 2004; Norby, 2011; Huntingford et al., 2013). This increase occurs despite a notable reduction in rainfall (-7.1% total) in central and eastern parts of Amazonia, which was not observable in the comparison with SSP585. Drying is often associated with increased risk of dieback in the Amazon forest (Parry et al., 2022), so the increase in both NPP and land carbon storage, despite decreases in precipitation is interesting. This is likely due to increased water use efficiency at higher concentrations of CO<sub>2</sub>, which is a strong compensating effect (Dekker et al., 2016).

The increase in NPP and land carbon storage is observed in all models except IPSL-CM6A-LR (Figure 17 and Figure 14), where there are small reductions in both, likely resulting from the strong drying trend in northern Amazonia in this model (Figure 13) (Flores et al., 2024). CESM2-WACCM experiences the largest increase in land carbon storage within Amazonia by a large margin, with an average increase of 3.84 kgC m<sup>-2</sup> (Figure 17). These increases are concentrated in central and western areas of Amazonia, with the furthest eastern region experiencing little to no increase in land carbon storage. This further suggests that CESM2-WACCM has a particularly strong CO<sub>2</sub> fertilisation effect compared to other models, though its increases in land carbon globally are concentrated around the equator. Interestingly, CNRM-ESM2-1, which experiences the largest increase in land carbon storage globally, shows relatively little increase in land carbon storage in Amazonia compared to other models, only IPSL-CM6A-LR has smaller increases.

Figure 18 shows ensemble mean variables against global warming and CO<sub>2</sub> concentrations. In G6sulfur, NPP exhibits a steeper increase with global warming compared to SSP245 and SSP585, which both track similar trajectories against temperature (Figure 18a). Plotting NPP against CO<sub>2</sub> shows overlapping trajectories across all three experiments, increasing steeply from pre-industrial levels before plateauing at around 800ppm (Figure 18b). The CO<sub>2</sub> fertilisation effect plays a crucial role in these global mean responses, as has been shown in previous studies (Jones et al., 2013; Cox et al., 2004; Norby, 2011; Huntingford et al., 2013). Elevated CO<sub>2</sub> levels in the SSP585 scenario yield a nearly 40% increase in NPP compared to SSP245

**Figure 14.** Maps showing the difference between the 2090-2100 means of G6sulfur and SSP245 for net primary productivity (kgC m<sup>-2</sup> yr<sup>-1</sup>) for individual models. The numbers shown in the titles give the global land average. Hatching indicates where differences in the forest fraction as prescribed by the land use scenarios between SSP245 and SSP585 is greater than 0.1 (see Figure S2).

**Figure 15.** Maps showing the difference between the 2090-2100 means of G6sulfur and SSP245 for land carbon (kgC m<sup>-2</sup>) for individual models. The numbers shown in the titles give the global land average. Hatching indicates where differences in the forest fraction as prescribed by the land use scenarios between SSP245 and SSP585 is greater than 0.1 (see Figure S2).

**Figure 16.** Maps showing the difference between the 2090-2100 means of G6sulfur and SSP245 in the Amazon for (a) surface temperature (°C), (b) precipitation (mm day<sup>-1</sup>), (c) net primary productivity (kgC m<sup>-2</sup> yr<sup>-1</sup>) and (d) land carbon (kgC m<sup>-2</sup>). The numbers shown in the titles give the average of the displayed area (a, b) or land average of the displayed area for each map (c, d). Stippling indicates regions where the standard deviation across the models is more than the ensemble mean change while hatching indicates where differences in forest fraction as prescribed by the land use scenarios between SSP245 and SSP585 is greater than 0.1 (see Figure S2).

(Figure 18b). Conversely, global NPP shows minimal sensitivity to temperature, demonstrated in Figure 18b where the change in total NPP in G6sulfur tracks SSP585 despite their different global mean temperatures, as a result of counteracting negative effects of warming in the tropics and positive effects in the high latitudes. G6sulfur and SSP585 show comparable global NPP increases relative to pre-industrial levels, consistent with global NPP being mainly dependent on CO<sub>2</sub> concentrations. However, projected global land carbon storage increases are larger in G6sulfur than either SSP245 or SSP585 (Figure 18c&d), but for different reasons.

Compared to SSP245, global land carbon storage is increased in G6sulfur due to extra CO<sub>2</sub> fertilisation (Figure 18c). This depends on the uncertain magnitude of CO<sub>2</sub> fertilisation of photosynthesis, although it remains clearly visible even in the CMIP6 models that include nitrogen limitations. Meanwhile, global land carbon storage is increased in G6sulfur compared to SSP585 due to reduced soil respiration as a result of the reduced warming in G6sulfur (Figure 18d). These results build upon the findings of single model analysis on the impacts of SRM on land carbon storage and NPP (Yang et al., 2020; Tjiputra

285

**Figure 17.** Maps for individual models (**a-e**) or the model ensemble mean (**f**) showing the difference between the 2090-2100 means of G6sulfur and SSP245 for land carbon storage (kgC m<sup>-2</sup>) in Amazonia. The numbers shown in the titles give the regional land average for each map. Stippling (**f**) indicates regions where the standard deviation across the models is more than the ensemble mean change, i.e. the coefficient of variation is more than 1, while hatching indicates where differences in the forest fraction as prescribed by the land use scenarios between SSP245 and SSP585 is greater than 0.1 (see Figure S3).

et al., 2016; Zhao and Cao, 2022; Duan et al., 2020), as well as analysis performed for idealised scenarios in which the solar constant is reduced (Jones et al., 2013). As far as global land carbon storage is concerned, G6sulfur therefore has the benefit of higher CO<sub>2</sub> for photosynthesis (as in SSP585), but without the counteracting negative impact of additional warming compared to SSP245. The increases in global land carbon storage in G6sulfur relative to SSP585 occur despite a global decrease in precipitation and shifts in regional rainfall patterns within G6sulfur. Increased water use efficiency at higher concentrations of CO<sub>2</sub>, likely compensates for a large part of the reduction in rainfall (Dekker et al., 2016). However, it is important to note that a reduction in precipitation has other impacts which, though not observable in NPP or land carbon storage, remain important to consider.

The ESMs analysed in this study do not fully represent all processes important to the carbon cycle. They do not, for instance, model the impact of topographical variation on groundwater and its impact on ecosystem productivity (Costa et al., 2023). These models also do not explicitly account for diffuse radiation and its impacts on the biosphere (Mercado et al., 2009; Chakraborty et al., 2022). However, we note that diffuse radiation fertilisation is expected to further increase land carbon storage under SAI. We also note that this study only analyses a small number of models for one SRM scenario, stratospheric

Figure 18. Evolution of the global mean changes in NPP (a, b) and land carbon storage (c, d) anomalies relative to the pre-industrial period (1850-1900) against temperature anomaly (a, c) and  $CO_2(b, d)$ .

aerosol injection, and that the risks and benefits of SRM depend strongly on how it is deployed (Haywood et al., 2022b; Keith and MacMartin, 2015). However, we see similar increases in projected NPP and land carbon storage in the G6solar experiments, which apply a reduction of the solar constant (Supplementary Figures S3&S4).

## 4 Conclusions

There are many legitimate concerns about the possibility of implementing SRM geoengineering in the real world (Robock, 2020; Abiodun et al., 2021; Pope et al., 2012; Trisos et al., 2018; Barrett, 2008; Keith et al., 2016), which need to be openly discussed. However, based on results from five Earth System Models, this study suggests that Stratospheric Aerosol Injection (SAI) geoengineering would likely increase global NPP and land carbon storage relative to both unmitigated climate change (under the SSP585 scenario) and conventional mitigation (as represented by SSP245 relative to SSP585). The modelled positive impacts of SAI are most marked in Amazonia, where SAI is projected to lead to significant increases in both NPP and land carbon storage. The best protection for the Amazon rainforest is a combination of reduced rates of both deforestation and anthropognenic climate change. However, this study suggests that SAI geoengineering might provide some emergency protection against climate-change induced Amazon carbon loss, if CO<sub>2</sub> induced climate change is not brought under control. In our view, further modelling studies are urgently needed to assess SRM approaches in that context.

Code availability. The code used for data analysis and producing the figures is available at https://doi.org/10.5281/zenodo.11507510 (Parry 305 et al., 2024). Please refer to the README file at Parry et al., 2024 for a detailed description of the code's functionality.

Data availability. The CMIP6 model output datasets analysed during this study are publicly available online at: https://doi.org/10.22033/ESGF/CMIP6.10 (Danabasoglu, 2019), https://doi.org/10.22033/ESGF/CMIP6.3907 (Séférian, 2019), https://doi.org/10.22033/ESGF/CMIP6.5059 (Boucher et al., 2020), https://doi.org/10.22033/ESGF/CMIP6.6448 (Niemeier et al., 2019), https://doi.org/10.22033/ESGF/CMIP6.5822 (Jones, 2019), https://doi.org/10.22033/ESGF/input4MIPs.10468 (Hurtt et al., 2019).

Author contributions. I.M.P. led the analysis and drafted the manuscript with support from P.D.L.R. and P.M.C.. P.M.C. and P.D.L.R. conceived of the research idea, and I.M.P., P.D.L.R., P.M.C. and J.M.H. shaped the research. All co-authors commented on and provided edits to the original manuscript.

Competing interests. The contact author has declared that none of the authors has any competing interests.

Acknowledgements. We acknowledge the World Climate Research Programme's Working Group on Coupled Modelling, which is responsible for CMIP, and we thank the climate modelling groups for producing and making available their model output. The IPSL-CM6 experiments
were performed using the HPC resources of TGCC under the allocations 2021-A0100107732, 2022-A0120107732, 2023-A0140107732
(project gencmip6) provided by GENCI (Grand Equipement National de Calcul Intensif).I.M.P, P.M.C, and J.M.H were part funded by
Quadrature Climate Foundation; grant reference number 01-21-000336. P.D.L.R and P.M.C were supported by the Optimal High Resolution
Earth System Models for Exploring Future Climate Changes (OptimESM) project, grant agreement number 101081193, and by ClimTip.

The ClimTip project received funding from the European Union's Horizon Europe research and innovation programme under grant no.
101137601, funded by the European Union. P.M.C was supported by the PREDICT project, which received funding from the European
Space Agency (ESA) under contract no. 4000146344/24/I-LR. P.D.L.R and P.M.C acknowledge support from the UK Advanced Research
and Invention Agency (ARIA) via project "AdvanTip", grant no. SCOP-PR01-P003.

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
