# Peer review of "Solar Radiation Modification is projected to increase land carbon storage and to protect the Amazon rainforest"

_EGUsphere, 2025_

## Referee Comment (RC2)

**Referee report on ESDD paper titled: "Solar Radiation Modification is projected to increase land carbon storage and to protect the Amazon rainforest"**

*Summary and Scope*

This climate multi-model intercomparison study aims to understand how climate intervention using stratospheric aerosol injection method can alter the land carbon storage (LCS) and net primary productivity (NPP), with a special focus on the Amazon region. The authors systematically discuss the differences in the simulated quantities of interest such as surface temperature, precipitation, NPP and LCS, and explain the reasons for those simulated differences.

The climate intervention scenario considered in this study (G6sulfur) aims to restore the global climate change induced by a fossil fueled development scenario (SSP585) to a middle of the road scenario (SSP245). The authors discuss both the exclusive effect of climate engineering (G6sulfur – SSP585) and the cumulative effect of restoring the climate (G6sulfur – SSP245). One of the major takeaways from this study is the enhanced NPP and LCS over the Amazon regions due to G6sulfur compared to both SSP585 and SSP245 across most models considered. The main reasons for this are due to larger precipitation, $CO_2$ fertilization effect (when comparing with SSP245) and restoration of surface temperature to an optimum value for photosynthesis, resulting from climate intervention, thereby leading to enhanced productivity in the Amazon region.

The manuscript structure and sequence of discussion is adequate but requires several changes to improve its readability.

*Explicit recommendation*

I recommend several **minor revisions** to the text before accepting.

*Detailed comments*

**Main comments**

1) While the apparent 'benefit' of SRM on NPP over Amazon has been highlighted in this study, the possible 'damage' and associated modelling uncertainty over other ecologically sensitive regions such as central Africa and SE Asia must also be emphasized. I recommend this to be done both in the abstract and the conclusions section. This is to avoid any possible miscommunication that SRM is only purely

beneficial for the Amazon region's productivity, without any side effects for other regions.

2) Structure section 3 (results) into smaller subsections. For example, exclusive effects (G6sulfur – SSP585) can become subsection 3.1, and the cumulative effects (G6sulfur – SSP245) can become subsection 3.2. You could also further divide it into specific parts where focused discussion related to Amazon region is present. This is to improve the readability and ease of navigating the results section.

3) Briefly summarize the exclusive climate change signal (SSP585 – SSP245): how does the temperature, precipitation, NPP and LCS change due to lack of mitigation measures. The discussion could then focus on how well or to what extent does G6sulfur restore/counteract this climate change signal.

4) Conclusions section: It would be good to have a brief quantitative summary of the main results of this study, in this section.

**Minor Comments**

L18: "little more than a decade more". Remove the repeated 'more'

L51: Aims to reduce forcing from SSP585 to SSP245? or global mean temperature? or both? Please clarify it here and make it consistent across the manuscript.

L134-135: "hatching in these plots removes changes in land carbon storage that are not due to SAI from our analysis.". This is inconsistent with the caption in Figure 4: hatching indicates where the forest fraction difference between SSP585 and SSP245 is more than 0.1.

L143: "warming of up to 2 K in some regions". Since this comparison is the exclusive effect of SRM, I wonder why it could induce additional warming to an already warm climate. Could be worth elaborating on.

Figure 4 caption: "land-use scenarios, differs by more than 0.1" has been written twice. Remove the repetition.

L182: "(a decrease of more than 1 mm/day". Close the parenthesis.

L182: Any explanation for why models simulate a decrease over north-western and increase over north-eastern South America? (both precipitation and NPP)

Figure 8: Please write "land carbon storage" instead of only "land carbon"

L197-199: The sentence is too long and confusing. Do you mean to run SRM simulations to simulate dieback in the absence of SRM?

L201: It is interesting and useful to note that all the models considered here have consistently/robustly simulated NPP and LCS increase over eastern Amazon. What can we say about the consistency in the simulated signal among the models over other regions?

Figures 4, 10, 11, 16: Please mention in first sentence of their captions that the model ensemble mean is plotted in these figures. You could also consider mentioning the figure numbers in which the quantities simulated by the individual models are plotted.

L224-L226: Long and confusing sentence. Large decreases and large increases both similarly found in the band around equator?

L231-L232: Cite Figs. 11d and 4d for readers' convenience.

L242: Fig. 14 shows NPP and not LCS. Why did you cite it here when NPP has not been discussed in this sentence?

L259: IPSL model shows decreases rather than smaller increases over most Amazon region. See Figure 17c.

L266: sensitivity to temperature is shown in 18a not 18b.

L268: could be better when a map is cited to illustrate the "warming in the tropics and positive effects in the high latitudes."

L271: These reasons are not clear from reading the next paragraph. In what way, to what extent and for what reasons is the LCS and NPP response different in G6S compared to SSP? Please explain concisely, preferably in the form of a table.

L291: Kalidindi et al., 2015 would be a good article to cite when discussing how different the primary productivity responses are to solar constant reduction and sulfate aerosol geoengineering.

Kalidindi, Sirisha, Govindasamy Bala, Angshuman Modak, and Ken Caldeira. "Modeling of solar radiation management: a comparison of simulations using reduced solar constant and stratospheric sulphate aerosols." *Climate Dynamics* 44, no. 9 (2015): 2909-2925.